# Seroprevalence of SARS-CoV-2 Antibodies in Employees of Three Hospitals of a Secondary Care Hospital Network in Germany and an Associated Fire Brigade: Results of a Repeated Cross-Sectional Surveillance Study Over 1 Year

**DOI:** 10.3390/ijerph19042402

**Published:** 2022-02-19

**Authors:** Anke Hildebrandt, Oktay Hökelekli, Lutz Uflacker, Henrik Rudolf, Michael Paulussen, Sören G. Gatermann

**Affiliations:** 1St. Vincenz Hospital, Department of Internal Medicine I, 45711 Datteln, Germany; o.hoekelekli@vincenz-datteln.de (O.H.); l.uflacker@vincenz-datteln.de (L.U.); 2Institute of Medical Microbiology, University Hospital Münster, 48149 Münster, Germany; 3Institute for Biostatistics and Informatics in Medicine and Ageing Research, University Medical Center Rostock, 18057 Rostock, Germany; henrik.rudolf@uni-rostock.de; 4Vestische Kinder- und Jugendklinik, Witten/Herdecke University, 45711 Datteln, Germany; m.paulussen@kinderklinik-datteln.de; 5National Reference Center for Multidrug-Resistant Gram-Negative Bacteria, Ruhr University Bochum, 44801 Bochum, Germany; soeren.gatermann@rub.de

**Keywords:** SARS-CoV-2, coronavirus, COVID-19, antibodies, healthcare workers

## Abstract

Healthcare workers (HCWs) are playing a vital role in the current SARS-CoV-2 pandemic. This study investigated how infection spreads within three local hospitals and an associated fire brigade in Germany by testing employees for the presence of SARS-CoV-2 IgG antibodies over one year. The three observational periods corresponded to the initial three pandemic waves: first wave: June–September 2020, second wave: October 2020–January 2021, and third wave: February–June 2021. We analysed 3285 serum samples of 1842 employees, which represents 65.7% of all employees. Altogether, 13.2% employees were seropositive: 194/1411 HCWs (13.7%) and 49/431 non-HCWs (11.4%) with a clear increase of seroprevalence from the first (1.1%) to the second (13.2%) and third (29.3%) pandemic wave. HCWs presumably had an additional occupational risk for infection in the second and third wave due to an increase of infection pressure with more COVID-19 patients treated, showing possible weak points in the recommended infection prevention strategy.

## 1. Introduction

Severe acute respiratory syndrome coronavirus 2 (SARS-CoV-2) is a novel beta coronavirus that was first identified in December 2019 in Wuhan, China [1,2], and became pandemic [3,4]. The WHO declared a global health emergency on 31 January, 2020; subsequently, on 11 March, 2020, they declared it a pandemic [5].

SARS-CoV-2 infection presents clinically as coronavirus disease 2019 (COVID-19) with a broad range of symptoms [6,7].

The current SARS-CoV-2 pandemic is a worldwide challenge for the medical sector. Healthcare workers (HCWs) are at specific risk for SARS-CoV-2 [8], especially if they are inadequately protected [9,10]. Serological testing of specific antibodies against SARS-CoV-2 has commonly been used to investigate infections of HCWs [11]. An average seroprevalence rate of 8% [12] and 8.6% [13] in HCWs were reported worldwide before the era of vaccination. Since January 2021, the possibility of vaccination has become an add-on to the personal protection and infection control measures.

Data from German HCWs are available from a variety of hospitals, but nearly all published data focus on the first pandemic wave [14,15,16,17,18,19,20] (Table 1). Two hospitals reported data until December 2020 [21,22], but no information for HCWs in Germany is available for the year 2021 so far.

Our study reports the course of seroprevalence of IgG antibodies against SARS-CoV-2 among employees of three local hospitals of a secondary care hospital network in Germany and an associated fire brigade, an institution mainly responsible for patient transport to and between hospitals, over 1 year (June 2020–June 2021). In addition, we evaluate if HCWs had an additional occupational risk for SARS-CoV-2 infection overtime. Finally, the results are interpreted in the epidemiological context of the local incidence, respectively.

## 2. Methods

### 2.1. Study Design

The study was a single center study conducted at the St. Vincenz Hospital Datteln (VHD) with 316 beds and the main departments: internal medicine, surgery, gynaecology, obstetrics, and urology. We looked separately at three time periods according to the pandemic waves: first wave: June–September 2020, second wave: October 2020–January 2021, and third wave: February–June 2021. The VHD belongs to the Vestische Caritas Kliniken GmbH hospital network. Within the study time of one year, two other hospitals of the secondary care hospital network took part: St. Laurentius Stift Waltrop (LSW) with 172 beds (geriatric and psychiatric department), and the children’s hospital Vestische Kinder- und Jugendklinik (VKJ) Datteln with 244 beds. An associated fire brigade, as institution mainly responsible for patient transport to and between hospitals, took part in the second and third pandemic wave. Furthermore, we investigated 40 employees of associated residential care homes for the elderly.

### 2.2. Enrolment and Data Management

All employees were invited to take part. Written informed consent included a questionnaire and agreement on providing a blood sample (not exceeding 9 mL of venous blood).

In the questionnaire personal data and questions for clinical symptoms 2 months prior testing were recorded. Additionally, we asked for exposure to confirmed COVID-19 cases, results of previous polymerase chain reaction (PCR), or previous serology. In January 2021, the question for COVID-immunization status was added.

Pseudonymized blood samples were sent to our central laboratory for testing of antibodies against SARS-CoV-2. Data from pseudonymized questionnaires were collected and processed with MS Excel 2010 (Microsoft Corporation, Redmond, WC, USA).

### 2.3. SARS-CoV-2 Antibody Testing

Presence of SARS-CoV-2 antibodies was investigated with the chemiluminescence-based immunoassay Elecsys, Anti-SARS-CoV-2 (Roche, Basel, Switzerland). The immunoassay targets recombinant nucleocapsid protein and was carried out according to manufacturer’s instructions. Sensitivity and specificity as provided by the manufacturer was high (≥99%) after 14 days post PCR confirmation. Participants with positive test results were regarded as SARS-CoV-2 seropositive. Re-testing was offered to all participants during the test period.

Available vaccines did not interfere with the SARS-CoV-2 antibody test we used in the study as they do not contain parts the nucleocapsid protein, but parts of the spike protein.

### 2.4. Outcomes

The primary aim of the study was to assess the course of seroprevalence of SARS-CoV-2 antibodies in hospital employees and an associated fire brigade during the first pandemic year using an IgG detecting immunoassay. Preliminary results of the first observation period are already published [39].

Secondary aims were:
(i)The detection of possible differences among.
(a)HCWs working in high-risk areas with regular contact to COVID-19 positive patients: COVID-19 ward, intensive care unit (ICU), and emergency department.(b)HCWs working in intermediate-risk areas with contact to COVID-19 negative patients.(c)Non-HCWs working in low-risk areas with no contact to patients at all (personnel working in administration, kitchen, cleaning service, and others).(d)Non-HCWs working in the fire brigade with intermediate-risk.(ii)The description of clinical symptoms 2 months before testing.(iii)Correlation of the results with the number of COVID-19 positive patients treated, and to the local incidence.

### 2.5. Statistical Analysis

In descriptive analyses, participant’s demographics, professions, symptoms, and other attributes of COVID-19 exposure were determined and compared for the whole cohort, and stratified by antibody test result, using absolute and relative frequencies. Clinical characteristics and test results were compared by Pearson’s chi-squared test or Fisher’s exact test. Risk factors for a positive antibody test result were estimated by univariate logistic regression, giving odds ratios and 95% confidence intervals versus the reference level for each main category of the characteristics. We applied a significance level of 0.05. Further, to assess influence of number of contacts also inside hospitals, we conducted a bivariable logistic regression with the two predictor variables institution (VHD, LSW, and VKJ) and profession. Data were analysed with the statistical software R [40].

## 3. Results

### 3.1. Characteristics of the Study Participants

Overall, from June 2020 to June 2021, 1842 of 2804 employees (65.7%) of the three hospitals VHD, LSW and VKJ with together 732 beds and an associated fire brigade took part in the study. We investigated 80.6% (1177/1460) of the employees of the two hospitals with regular adult care (VHD and LSW) and 41.8% (522/1250) of employees of the children’s hospital VKJ. Additionally, 99.0% (103/104) employees of an associated fire brigade took part in the study. Baseline characteristics, such as age, profession, and working area for the observation period are shown in Table 2. The 40 participated employees of associated residential care homes for the elderly are included in Table 2 and Table 3, but not considered separately in the results and discussion section.

### 3.2. Seroprevalence

#### 3.2.1. First Observational Period: June–September 2020

From June 2020 to the end of September 2020, 14 of 1241 participants (1.1%) were seropositive: 5/674 (0.7%) in VHD, 8/235 (3.4%) in LSW, and 1/300 (0.3%) in VKJ. The fire brigade did not take part in the first pandemic wave.

#### 3.2.2. Second Observational Period: October 2020–January 2021

From October 2020 to the end of January 2021, 137 of 1035 participants (13.2%) were seropositive: 72/395 (18.2%) in VHD, 42/231 (18.2%) in LSW, 19/356 (5.3%) in VKJ, and 2/46 (4.3%) in the fire brigade.

#### 3.2.3. Third Observational Period: February–June 2021

From February 2021 to the end of June 2021, 209 of 714 participants (29.3%) had detectable antibodies: 116/307 (37.8%) in VHD, 48/108 (44.4%) in LSW, 32/240 (13.3%) in VKJ, and 11/57 (19.3%) in the fire brigade.

#### 3.2.4. Total Seroprevalence within the Year of Observation

Over the observational period of one year, the seroprevalence rate against SARS-CoV-2 increased continuously from 1.1% in the first, to 13.2% in the second, and to 29.3% in the third pandemic wave, respectively. Altogether, 360 of 2990 tests (12.0%) were seropositive in the three observational periods, including multiple tests of employees from wave to wave. In Figure 1 the number of already known seropositive employees and newly diagnosed seropositive employees is illustrated for the three time periods. Taking multiple tests out, 243/1841 employees were tested seropositive at least once within the year of observation which represents a total seroprevalence of 13.2%. There were differences between the four working areas (hospitals and fire brigade): 133/858 of employees (15.5%) were positive in VHD, 59/319 (18.5%) in LSW, 35/522 (6.7%) in VKJ, and 13/103 (12.6%) in an associated fire brigade, respectively.

### 3.3. Seroprevalence Associated with Age

We categorised three age groups: 16–25 years (*n* = 301), 26–40 years (*n* = 527), and >40 years (*n* = 999) (Table 2). The background for the classification into these groups was the assumption that participants might have different composition of their households (e.g., <25 years: less children, 26–40 years: young children, >40 years: older children) and consequently different risks for acquiring SARS-CoV-2 infection outside the hospital. In our statistical analysis we saw a significant lower risk of infection in the group >40 years (OR 0.65, 95% CI 0.46; 0.94) (Table 2) and, if we look at the three different observation periods separately, in the third pandemic wave (OR 0.59, 95% CI 0.37; 0.95) (Appendix A).

Following our hypothesis, employees in this age group perhaps had no or older children living in their households than the youngest age group resulting in less contacts. Furthermore, children were tested regularly in schools while in preschools tests were voluntary resulting perhaps in more infection control especially in the third pandemic wave. However, we unfortunately did not collect data on household composition.

### 3.4. Seroprevalence Associated with Risk at Work

#### 3.4.1. Intermediate-Risk and High-Risk HCWs 

Altogether, 194 of 1411 tested HCWs (13.7%) were seropositive: 152/1223 intermediate-risk HCWs (12.4%), working with non-COVID-19 patients, and 42/188 high-risk HCWs (22.3%) working on the COVID-19 ward, ICU and emergency department. Looking at the three pandemic waves, we saw a significant higher risk of infection in both groups of HCWs compared to low-risk non-HCWs (Table 2).

#### 3.4.2. Low-Risk and Intermediate-Risk Non-HCWs

Altogether, 49 of 431 tested non-HCWs (11.4%) were seropositive: 36/328 employees (11.0%) working in low-risk areas with no contact to patients at all, and 13/103 employees (12.6%) working in the fire brigade with intermediate-risk while taking care of patients during transports (Table 2).

#### 3.4.3. Risk According to Profession and Institution

Employees of the two hospitals of adult care (VHD and LSW) had SARS-CoV-2 infections in employees working regularly with patients (MDs, nurses, care workers, therapists) and working without patients, summarised as other professions (e.g., kitchen, administration, cleaning service). In the children’s hospital (VKJ) employees with no contact to patients had no SARS-CoV-2 IgG antibodies in our study (Appendix A). The differences in employees working with patients compared to others was statistically significant especially for nurses (OR 1.64, 95% CI 1.09; 2.55) and care workers (OR 2.07, 95% CI 1.21; 3.54) (Appendix A). Additionally to the profession, employees in the two hospitals of adult care had a significant higher risk of SARS-CoV-2 infection compared to employees of the children’s hospital (Figure 2).

### 3.5. Clinical Symptoms of Seropositive Employees and Available PCR Results 

Independent from this study, employees were tested with PCR according to the recommendations of the Robert Koch Institute (RKI) in case of cold-like symptoms of any severity, exposure to COVID-19 positive persons, and returning from a region at risk [41]. Analysing available information about positive PCR results we identified SARS-CoV-2 infections in employees where both PCR and IgG antibodies were positive, in subjects where PCR was positive with no detectable IgG antibodies, and in subjects with positive PCR with no information about detectable antibodies. Altogether, 93.4% seropositive employees (227/243) had a positive PCR test previously, according to the test criteria of the RKI, and 6.6% seropositive employees (16/243) were missed with this PCR test strategy [41] (Table 3).

Seropositive employees had significantly more frequently clinical symptoms within the last 2 months prior the test. We found a high correlation especially with the symptoms, taste/smell disorders (OR 20.5, 95% CI 13.2–32.4), fever (OR 5.82, 95% CI 3.83; 8.81), and headache (OR 4.47, 95% CI 3.16–6.29) (Table 2).

### 3.6. SARS-CoV-2 IgG Antibody Titer

During the time of observation we got exactly one positive SARS-CoV-2 IgG antibody titer of 94 employees (Figure 3a), and follow-up titers in 149 seropositive employees (Figure 3b).

### 3.7. Seroprevalence with Regard to the Treated COVID-19 Patients and the Epidemiological Context

As the incidence of COVID-19 increased in the general population in the region of Recklinghausen, the number of COVID-19 patients increased in the hospitals. There was especially an increase of infection pressure in the second and third wave, and seroprevalence in employees increased accordingly (Figure 1 and Figure 4). The SARS-CoV-2 restrictions measures in the county Recklinghausen followed the nationwide regulations (https://www.bundesgesundheitsministerium.de/coronavirus/chronik-coronavirus.html). Basic points of these restrictions are summarized in Figure 5. Altogether, 552 COVID-19 patients were treated in the three hospitals (VHD: 380, LSW: 106, VKJ: 66) during the observation period. Interestingly, the LSW had no COVID-19 positive patient in the first infection period (Figure 4). Hence, the eight seropositive HCWs in this hospital observed in the first wave presumably acquired their infection not nosocomial, but more likely during private contacts.

## 4. Discussion

Looking at the risk to acquire the SARS-CoV-2 infection privately or at work, we may assume that the participating hospitals were probably no places of infectious spread to employees in the first pandemic wave. Some studies on HCWs in Germany also differentiated between seroprevalence rates according to the risk at work [14,21], and investigated high-risk group HCWs on ICU, the COVID-19 ward and emergency department [19,22,23,31]. All these studies found a low seroprevalence up to 4.36% [18] in the first pandemic wave. This is consistent to our results.

Only sparse seroprevalence data of HCWs are available for the second and third wave in Germany. Korth et al. found a seroprevalence rate of 5.1% in 315 HCW between August and December 2020 at the University Hospital Essen, close to our region [21]. Another study in Oberspreewald-Lausitz reported a seroprevalence of 13.3% in 166 HCWs between July and December 2020 in a standard care hospital [35]. During the second and third observational period in our study, the incidence in the general population of the region of Recklinghausen and the number of COVID-19 patients increased markedly. Accordingly, the seroprevalence in employees in our study went up sharply to 13.2% in the second, and to 29.3% in the third wave, respectively, with a significant increase especially in high-risk HCWs regularly working with COVID-19 positive patients. We saw infections spread on regular wards in adult care, even in the geriatric department, resulting also in infections of employees. On the COVID-19 ward in the VHD at that time there were several potential contributors to an additional occupational risk to get infected with SARS-CoV-2: e.g., structural alteration works on the ward with relocation to another ward, a high turnover of patients, and longer retention time of deceased patients on the ward due to a temporary lack of storage capacities. Additionally, infected high-risk HCWs led to personnel shortage with personnel shifting, high workload with less time for correct self-protection and a shortage of manpower to instruct new personnel carefully. Ongoing infections on regular wards in both hospitals of adult care (VHD and LSW) led to an additional occupational risk of infection in intermediate-risk HCWs. In contrast, due to the lower hospitalization rate of COVID-19 in children, no such effects were observed in the children’s hospital, nor were there transmissions between staff noticed. Data on nosocomial spread of infections within the two hospitals of adult care during the second and third waves were actually evaluated in a retrospective study to get more information about the weak points in the concept of infection prevention.

The population-based sequential study “MuSPAD” investigated the SARS-CoV-2 seroprevalence of the general not vaccinated population in seven regions in Germany between July 2020 and May 2021. The authors report a low seroprevalence of 1.3–2.8% after the first pandemic wave and an increase up to 4.1–13.1% until May 2021 [42]. From October 2020 to February 2021 a nationwide seroepidemiological study in Germany called “RKI-SOEP-Study” noticed a clear increased risk of SARS-CoV-2 infection for employees of healthcare professions (4.6%) compared to non-healthcare employees (1.8%) [43].

First SARS-CoV-2 seroprevalence data of extraclinical personnel depending on their operational area in the fight against the COVID-19 pandemic was recently reported from Brune et al. in Essen, Germany. The authors detected in 8 of 732 employees of the professional fire brigade and aid organizations in the city area SARS-CoV-2 IgG antibodies in the first pandemic wave which corresponds to a seroprevalence rate of 1.1% [37]. Our study investigated employees of the fire brigade in the second and third wave with a seroprevalence rate of 12.6% (13/103) according to the high incidence in the general population at that time. Unfortunately, we have no information about the number of transported COVID-19 positive patients. Because of the seroprevalence of 11.0% (36/328) in employees with no contact to patients, we expect that the fire brigade had no or at best a minimal additional occupational risk for infection in our study.

The longitudinal course of SARS-CoV-2 antibodies in HCWs after infection was similar to that reported by other authors who investigated infected patients, with the Spike IgG titers showing only modest declines at 6 to 8 months [44].

An important limitation of our study is its convenient sampling design. It is conceivable that employees with a previous COVID-19 diagnosis or symptoms consistent with COVID-19 would be more interested in a measurement of their serum COVID-19 antibodies and follow-up titers. Many employees wanted to know their antibody titer for the decision to get vaccinated. Those employees who were hesitant to get vaccinated, tent to take part in the study repeatedly. There are several reasons for missing following antibody tests. The shortage of antigen tests and antibody tests at the beginning of the pandemic was overcome at the beginning of 2021, so that employees could also get tested ambulant by their general practitioner. For persons who got vaccinated (two vaccines or one vaccine 3 months after infection), the antibody test result and follow-up titer was not relevant any longer.

However, our study also has strengths. It presents data on the course of SARS-CoV-2 seroprevalence based on a cohort of employees of three hospitals and an associated fire brigade. The time period of one year contains a “pre-vaccination” period, and a short period after vaccination had started. We could not demonstrate an effect of the vaccination in this short “post-vaccination” observation period so far, but it will be interesting to investigate the seroprevalence of vaccinated employees again in the following pandemic waves.

## 5. Conclusions

The overall seroprevalence in the investigated employees was 13.2% after one year. Community acquired transmission seems to have played a larger role for SARS-CoV-2 infection than professional exposure during the first pandemic wave. However, this resulted from an overall low exposure of hospital employees to COVID-19 positive patients at a time where the region was not a SARS-CoV-2 hotspot. With the increase of infection pressure in the second and third pandemic wave, HCWs had an additional occupational risk for infection, and we observed some hints towards in-hospital transmission. This underlines the need to adapt the concept of infect prevention continuously, especially in situations with structural limitations, such as high workload and personnel shortage in a pandemic, in order to keep hospitals safe places.

## Figures and Tables

**Figure 1 ijerph-19-02402-f001:**
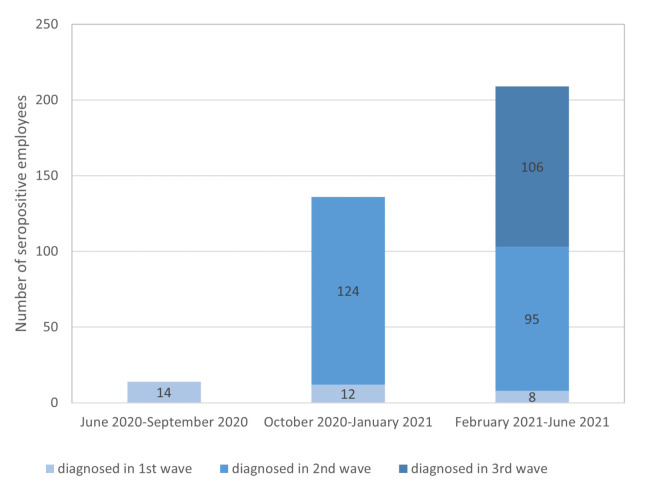
Seroprevalence of SARS-CoV-2 IgG antibodies from June 2020 to June 2021. Any employee in a given sector could volunteer to participate in the study at any time, so that we did not test all positive employees in the following observational period again (lack of follow-up).

**Figure 2 ijerph-19-02402-f002:**
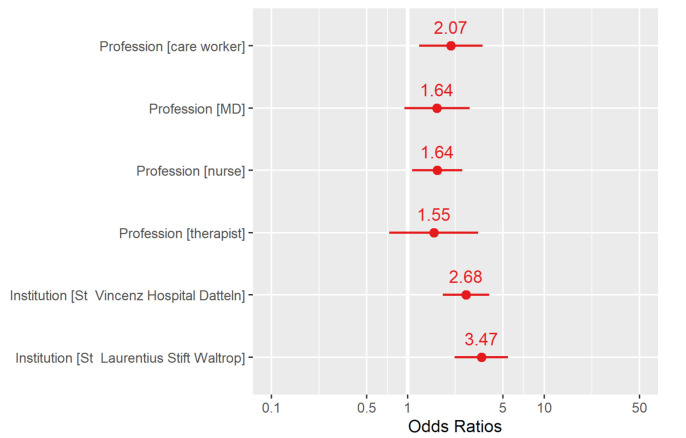
Risk of profession and institution for infection. To assess influence of number of contacts also inside hospitals, we conducted a bivari-able logistic regression with the two predictor variables institution (VHD, LSW, and VKJ) and profession. The latter was recoded regarding patient contact, specifying med-ical doctor, nurse, care worker, therapist, and all other professions without patient con-tact as “others”. To analyse if there were excess risk, the children’s hospital VKJ and “others” profession were chosen as reference. Results are displayed in a forest plot showing odds ratios.

**Figure 3 ijerph-19-02402-f003:**
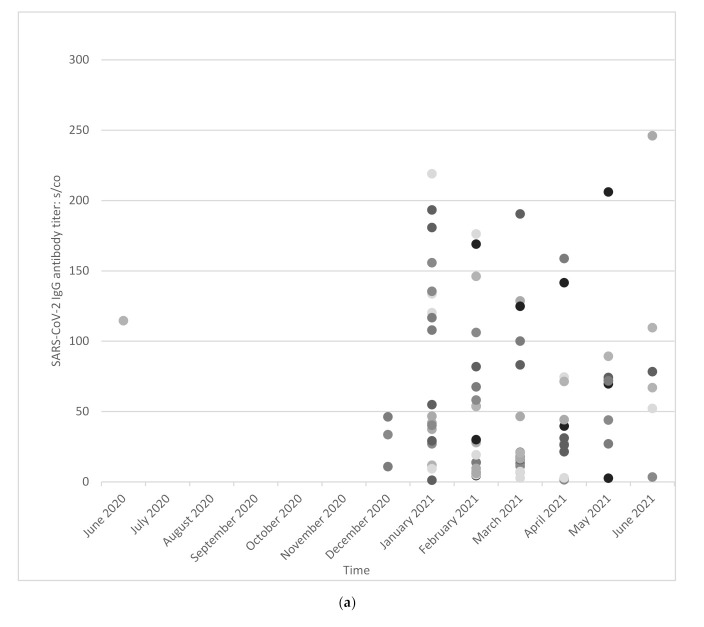
SARS-CoV-2 IgG antibody titer: signal to cut-off (s/co) index. (**a**) A total of 94 employees with exactly one positive blood sample. Each dot represents an employee who had exactly one positive antibody titer against SARS-CoV-2. (**b**) A total of 149 employees with more than one positive antibody titer against SARS-CoV-2. Each line represents one or more positive follow-up titers of one employee.

**Figure 4 ijerph-19-02402-f004:**
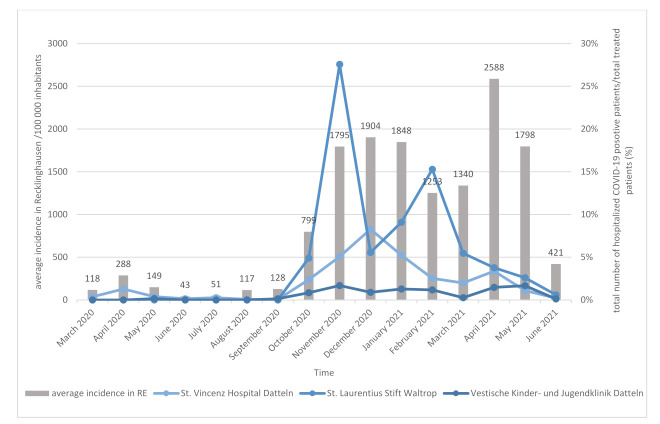
SARS-CoV-2 incidence in Recklinghausen (RE) and total number of hospitalized positive patients/total treated patients (%) from March 2020 to June 2021. Average incidence in Recklinghausen was provided by the interactive dashboard of the website: https://www.kreis-re.de (accessed on 25 December 2021).

**Figure 5 ijerph-19-02402-f005:**
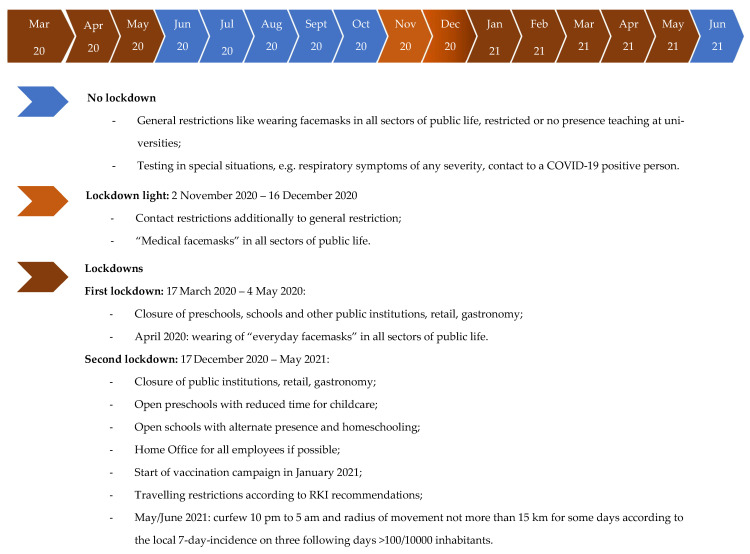
SARS-CoV-2 restriction measures in the county Recklinghausen followed the nationwide regulations (available online: https://www.bundesgesundheitsministerium.de/coronavirus/chronik-coronavirus.html (accessed on 30 January 2022).

**Table 1 ijerph-19-02402-t001:** Published SARS-CoV-2 seroprevalence data in HCWs in Germany until December 2021.

Hospital	Time Period	Number of Employees Tested	SARS-CoV-2 IgG Antibody Rate	Additional Information	Reference
University Hospital Bonn	First wave	217 frontline HCWs	1.86%		[23]
University Hospital Jena	First wave	660 employees	1.90%		[14]
University Regensburg	First wave	166 HCWs	0.00%	Perinatal center	[15]
BDH Clinic Hessisch Oldendorf	First wave	406 employees	2.70%	Neurological center	[24]
Saalfeld	First wave	45 employees, 20 HCWs	1.54%	Cleaning staff, oncological ward	[25]
University Hospital Essen	First wave, second wave	March–December 2020 450 HCWs	March–May 2020: 2.2%; June–July 2020: 4.0%; August–December 2020: 5.1%	Classification into high-risk, intermediate-risk and low-risk group	[21,26]
Weiden and Tirschenreuth, Bavaria	First wave (July 2020)	1838/2387 HCWs (77%)	15.1% HCWs	Region with highest rate of infection	[16]
University Hospital Munich	First wave	151 HCWs	2.60%		[27]
Altona Children’s hospital Hamburg	First wave	619 hospital employees	0.33%	70.3% of total staff	[17]
Hamburg	First wave	871 employees April 2020 406 employees follow-up in October 2020	4.36% 0.74%	Secondary care hospital	[18,28]
Heilbronn	First wave	3067 HCWs	3.50%	COVID-19 treatment center	[19]
Ulm	First wave	394 HCWs	0.25%	Residential care home for the elderly	[29]
University Medical Center Hamburg-Eppendorf	First wave	1253 employees including 1026 HCWs	1.80%		[20]
Munich	First wave	300 HCWs	4.67%	Quaternary care hospital	[30]
Saalfeld	First wave	68 HCWs	0%	ICU and COVID-19 ward	[31]
Two university hospitals in Brandenburg	First wave	1013 HCWs	2.1–2.2%	Ongoing study	[32]
Ortenaukreis	First wave	198 HCWs	3.50%	Regional medical center and several outpatient facilities	[33]
Fulda	First wave	1800 HCWs	1%		[34]
HCWs recruited in all parts of Germany	First wave	516 HCWs	3.50%	Intensive care and emergency care unit	[22]
Oberspreewald-Lausitz	First wave, second wave (July–December 2020)	166 HCWs	13.30%	Standard care hospital	[35]
Saalfeld	First wave	18 employees	0.00%	Regular ward	[36]
Essen	First wave	732 employees	1.1%	Professional fire brigade and aid organizations in the city area	[37]
Berlin	First wave	1477 HCWs 1223 HCWs	May/June 2020: 1.2% December 2020: 4.6%	Tertiary care hospital	[38]
St. Vincenz Hospital Datteln	First wave, second wave, third wave	1842 employees including 1411 HCWs	June–September 2020: 1.2% October 2020–January 2021: 13.2% February–June 2021: 29.3%	Employees of three hospitals of a secondary care hospital network and an associated fire brigade	[39], actual data

**Table 2 ijerph-19-02402-t002:** Characteristics of the study population—stratified by SARS-CoV-2 IgG antibody results.

Variable	SARS-CoV-2 Antibodies Total			Statistics	
	All Employees Without Repeated Testing			OR (95% CI)	*p*-Value
	All	Not Detectable	Detectable		
	*n* = 1842	*n* = 1599	*n* = 243		
**Age ^a^**					0.040
16–25 years	301 (16.5%)	251 (15.8%)	50 (20.7%)	Reference	
26–40 years	527 (28.8%)	450 (28.3%)	77 (31.8%)	0.86 [0.58; 1.27]	
>40 years	999 (54.7%)	884 (55.8%)	115 (47.5%)	0.65 [0.46; 0.94]	
**Sex**					0.396
Male	1503 (81.6%)	1310 (82.0%)	193 (79.4%)	Reference	
Female	339 (18.4%)	289 (18.0%)	50 (20.6%)	1.18 [0.83; 1.64]	
**Hospital/Institution**					<0.001
St. Vincenz Hospital Datteln	858 (46.6%)	725 (45.3%)	133 (54.7%)	Reference	
St.-Laurentius-Stift Waltrop	319 (17.3%)	260 (16.3%)	59 (24.3%)	1.24 [0.88; 1.73]	
Children’s Hospital Datteln	522 (28.3%)	487 (30.5%)	35 (14.4%)	0.39 [0.26; 0.57]	
Fire brigade	103 (5.6%)	90 (5.6%)	13 (5.4%)	0.80 [0.41; 1.42]	
others	40 (2.2%)	37 (2.3%)	3 (1.2%)	0.46 [0.11; 1.31]	
**Profession**					0.7135
Nurse	927 (50.3%)	802 (50.2%)	125 (51.4%)	Reference	
Medical doctor	215 (11.7%)	187 (11.7%)	28 (11.5%)	0.96 [0.61; 1.48]	
Care worker	201 (10.9%)	168 (10.5%)	33 (13.6%)	1.26 [0.82; 1.90]	
Cleaning service	36 (2.0%)	31 (1.9%)	5 (2.1%)	1.06 [0.35; 2.57]	
Administration staff	136 (7.4%)	124 (7.8%)	12 (4.9%)	0.63 [0.32; 1.13]	
Fire brigade	103 (5.6%)	90 (5.6%)	13 (5.4%)	0.94 [0.48; 1.67]	
Kitchen	35 (1.9%)	31 (1.9%)	4 (1.7%)	0.86 [0.25; 2.22]	
Therapist	88 (4.8%)	75 (4.7%)	13 (5.4%)	1.12 [0.58; 2.02]	
Other profession	101 (5.5%)	91 (5.7%)	10 (4.1%)	0.71 [0.34; 1.35]	
**Risk of COVID-19 infection**					<0.001
*non-HCW*					
Low-risk-group: working without patient contact	328 (17.8%)	292 (18.3%)	36 (14.8%)	Reference	
Intermediate-risk group: fire brigade	103 (5.6%)	90 (5.6%)	13 (5.4%)	1.18 [0.58; 2.28]	
*HCW*					
Intermediate-risk group	1223 (66.4%)	1071 (67.0%)	152 (62.6%)	1.15 [0.79; 1.71]	
High-risk group	188 (10.2%)	146 (9.1%)	42 (17.3%)	2.33 [1.43; 3.81]	
**Number of symptoms within the last 2 months**					<0.001
No symptoms	1373 (74.5%)	1246 (77.9%)	127 (52.3%)	Reference	
1 symptom	268 (14.5%)	230 (14.4%)	38 (15.6%)	1.62 [1.09; 2.38]	
2 symptoms	78 (4.2%)	60 (3.8%)	18 (7.4%)	2.96 [1.65; 5.08]	
3 symptoms	77 (4.2%)	48 (3.0%)	29 (11.9%)	5.93 [3.57; 9.69]	
4 symptoms	46 (2.5%)	15 (0.9%)	31 (12.8%)	20.1 [10.7; 39.3]	
5 symptoms					
**Clinical symptoms within the last 2 months ^b^**					
Cold-like symptoms	173 (9.4%)	126 (7.9%)	47 (19.3%)	2.81 [1.93; 4.03]	<0.001
Headache	179 (9.7%)	116 (7.3%)	63 (25.9%)	4.47 [3.16; 6.29]	<0.001
Fever	105 (5.7%)	60 (3.8%)	45 (18.5%)	5.82 [3.83; 8.81]	<0.001
Cough	261 (14.2%)	205 (12.8%)	56 (23.0%)	2.04 [1.45; 2.83]	<0.001
Hoarseness	17 (0.92%)	15 (0.94%)	2 (0.82%)	0.93 [0.13; 3.37]	1.000
Taste or smell disorders	104 (5.7%)	32 (2.0%)	72 (29.6%)	20.5 [13.2; 32.4]	<0.001
**Additional information**					
*Previous PCR testing ^c^*					<0.001
No information	1169 (63.5%)	1143 (71.5%)	26 (10.7%)	Reference	
PCR without known result	38 (2.1%)	24 (1.5%)	14 (5.8%)	25.4 [10.9; 58.2]	
Positive	173 (9.4%)	4 (0.3%)	169 (69.5%)	1715 [620; 8192]	
Negative	462 (25.1%)	428 (26.8%)	34 (14.0%)	3.49 [2.01; 6.13]	

Note: ^a^ date of birth is missing in 15 employees; ^b^ multiple answers possible; ^c^ employees were previously tested with PCR according to the R.K.I. recommendations [41].

**Table 3 ijerph-19-02402-t003:** Basic information on the conditions in participated institutions in general and with regard to SARS-CoV-2 infections.

Basic Information	St. Vincenz Hospital Datteln	St. Laurentius Stift Waltrop	Vestische Kinder -und Jugendklinik Datteln	Fire Brigade	Other Institutions	Total Number
Beds	316	172	244	n.a.	n.a.	n.a.
Employees	1085	375	1240	104	n.I.	2804
Tested employees (%)	862 (79.4%)	324 (86.4%)	528 (42.6%)	103 (99.04%)	40/n.I.	1817 (64.8%)
**SARS-CoV-2 IgG** **antibodies tested** **employees**						
June 2020 to September 2020: positive/tested (%)	5/674 (0.7%)	8/235 (3.4%)	1/300 (0.3%)	-	0/32	14/1241 (1.13%)
October 2020 to January 2021: positive/tested (%)	72/395 (18.2%)	42/231 (18.2%)	19/356 (5.3%)	2/46 (4.3%)	2/7 (28.6%)	137/1035 (13.2%)
February 2021 to June 2021: positive/tested (%)	116/307 (37.8%)	48/108 (44.4%)	32/240 (13.3%)	11/57 (19.3%)	2/2 (100%)	209/714 (29.3%)
**Altogether**						
Positive tests/tests	249/1522 (16.4%)	123/668 (18.4%)	68/951 (7.1%)	13/103 (12.6%)	4/41 (9.75%)	457/3285 (13.9%)
Positive employees/tested employees (%) (counting all only once in each wave)	193/1376 (14.0%)	98/574 (17.1%)	52/896 (5.8%)	13/103 (12.6%)	4/41 (9.75%)	360/2990 (12.0%)
Positive employees/tested employees (%) (counting each employee only once at all)	133/858 (15.5%)	59/319 (18.5%)	35/522 (6.7%)	13/103 (12.6%)	3/40 (7.5%)	243/1842 (13.2%)
vaccinated employees/tested employees (February 2021 to June 2021)	155/307 (50.5%)	43/108 (39.8%)	158/240 (65.8%)	22/57 (38.6%)	1/2 (50%)	
**COVID-19 patients**						
Hospitalised patients	380	106	66	n.a.	n.a.	
Patients on intensive care unit	36	n.a.	4	n.a.	n.a.	
Patients died	48	0	0	n.a.	n.a.	
Outpatients	82	n.a.	5 *	n.a.	n.a.	
**SARS-CoV-2 infection in employees**						
PCR positive	137	63	41	n.I.	n.I.	
PCR positive, IgG positive	133	59	35	n.I.	n.I.	
PCR positive, IgG antibodies negative	4	4	n.I.	n.I.	n.I.	
PCR positive, IgG not tested	28	12	n.I.	n.I.	n.I.	

**Note:** n.I.—no information; n.a.—not applicable; *—the correct number could not be counted because not all outpatients were tested in the whole observation period.

## Data Availability

The datasets and materials used and/or analysed during the current study are available from the corresponding author upon reasonable request.

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
