# Peer review of "Seroprevalence of SARS-CoV-2 Antibodies in Employees of Three Hospitals of a Secondary Care Hospital Network in Germany and an Associated Fire Brigade: Results of a Repeated Cross-Sectional Surveillance Study Over 1 Year"

_ijerph, 2022, doi:10.3390/ijerph19042402_

Round 1

Reviewer 1 Report

In this study, the authors present data on the seroprevalence for SARS-CoV-2 antibodies during three waves (1 year) of Covid-19 pandemic from three associated medical centers and a fire brigade from a German region. Seroprevalence increased gradually in each wave and personnel in closest contact with patients with Covid-19 appeared to be most affected. The effect of vaccination against Covid-19 was not examined.

The manuscript is of interest and well-written. The convenient sampling and lack of truly longitudinal data for all study participants are the major drawbacks. There are some minor issues that the authors could address to improve their manuscript:

  1. Introduction, aim of the study, page 2. The authors could briefly explain why a fire brigade was of interest to the study. It only becomes apparent later in the Methods.
  2. Section 2.4, page 4. The authors could mention if there is cross-reactivity with vaccine-generated antibodies.
  3. Results, section 3.1, page 5: Were individuals who did not consent to participate had different characteristics in terms of SARS-CoV-2 positivity predictors from those who did?
  4. Discussion, limitations, page 12. Could the authors discuss how robust their findings are in terms of the study design limitation? Could also the opposite happen, i.e., personnel who was once positive may have missed testing or did not consent to participate?
  5. Figure 2a and b. It is not clear what the different shades of grey depict in the two panels. Could the authors explain in the figure caption?

Reviewer 2 Report

In this work, "Seroprevalence of SARS-CoV-2 antibodies in employees of three hospitals of a secondary care hospital network in Germany and an associated fire brigade: results of a repeated cross-sectional surveillance study over 1 year" the authors aim to examine the risk of contracting SARS-CoV-2 infection privately or at work by monitoring the serological status of hospital workers during the three waves of the COVID-19 pandemic in Germany (June 2020-June2021).

Three local hospitals and an associated fire brigade were included representing 3285 serum samples from 1842 employees (65.7% of all employees). 13.7% of healthcare workers (HCWs) and 11.4% of non-HCWs were seropositive with an increase of seroprevalence according to the risk area of the employees and successive waves.

The authors conclude that during the 1st pandemic wave, private transmission played a larger role in the spread of infection, while during the 2nd and 3rd waves, HCWs had an additional occupational risk of infection with the increase in infectious pressure.

This study follows work published last year in the “International Journal of Hygiene and Environmental Health” concerning the anti-SARS-CoV-2 serological status of hospital employees in two German hospitals between June 2020 and September 2020.

This report is well written and interesting to investigate the spread of the infection among HCWs in Recklinghausen (Germany).

Minor issues:

L92: Sensitivity of the assay is ≥99% after 14 days post PCR confirmation

L105: “non-COVID-19 positive patients” means “negative COVID-19 patients“.

L120: Correct “0,05” by “0.05”.

L189: Correct “an” by “a” positive PCR test.

L211: The number of COVID-19 patients in LSW hospital are not the same in table 3 (102 in the text, 106 in table 3).

L212: “Interestingly, the LSW had no COVID-19 positive patient in the 1st infection period 212 (Table 3)”. Table 3 does not describe the number of patients during the different periods. Do the authors mean Figure 3?

Table 1 is a review of the literature regarding HCWs’ infection in Germany, and does not describe the work of the authors.

Figure 2: Caption is missing and Y axis titles are missing in both Fig2a and b.

Figure 3: Y axis titles are missing, legend is missing.

Table 3: Vaccinated employees (02-2021 to 06-2021): Is it the number of vaccinated people among all the employees or tested employees (e.g, St. Vincenz Hospital Datteln 155 vaccinated employees/1085 employees i.e., 14%)? Indicate % for all hospitals.

COVID-19 patients: Is it the number of patients for the three waves? What does 5+x means?

Figure 2a: All the first 243 positive serologies must be represented on this figure, not just 94.

Other concerns to discuss:

First, the limitations of the study. The authors acknowledge that there may be bias regarding the recruitment of the volunteers in their study. Other factors may influence the conclusion of this study.

What is the role of variants in the spread of infection among HCWs?

What are the restriction measures during the different waves in this region (e.g., lockdown, curfew, traffic restriction, wearing a mask outdoors...).

Table 2: Three age groups were categorized without explanation. How and why was the reference group chosen? Age is statistically associated with the proportion of positive serologies, the oldest being the least contaminated (16, 15 and 11.5% in 16-25, 26-40 and >40 years old, respectively. Do these proportions change during the different waves? What are the hypotheses about the observed differences in the risk of infection?

Table 2: Profession is not associated with the risk of infection, likely due to the inclusion of hospitals with no (or few) hospitalized COVID-19 patients. It would be interesting to show the risk associated with the profession in the different hospitals (and the evolution with the SARS-CoV-2 incidence.

How is the part of re-infection taken into account (with new variants or not)?

This is a German-Germanic study and the authors do not compare their results with other studies in Europe or worldwide.

Reviewer 3 Report

The article has the primary purpose of assessing the course of seroprevalence of sars-cov-2 antibodies in employees of 3 hospitals during a year of the pandemic and to investigate potential risk factors in Germany. The article reports basically descriptive results. Figures are not easy to read, it is better to eliminate them and report the results if necessary in the text. It is not clear how the incidences in Figure 3 were calculated. The article reports results of limited interest, roughly discussed.

Round 2

Reviewer 3 Report

The manuscript has improved compared to the first version, also in the reporting of the results